# Evaluation of Physico-Mechanical Properties on Oil Extracted Ground Coffee Waste Reinforced Polyethylene Composite

**DOI:** 10.3390/polym14214678

**Published:** 2022-11-02

**Authors:** Hoo Tien Nicholas Kuan, Ming Yee Tan, Mohamad Zaki Hassan, Mohamed Yusoff Mohd Zuhri

**Affiliations:** 1Department of Mechanical and Manufacturing Engineering, Faculty of Engineering, Universiti Malaysia Sarawak, Kota Samarahan 94300, Malaysia; 2Razak Faculty of Technology and Informatics, Universiti Teknologi Malaysia, Jalan Sultan Yahya Petra, Kuala Lumpur 54100, Malaysia; 3Advanced Engineering Materials and Composites Research Centre, Department of Mechanical and Manufacturing Engineering, Universiti Putra Malaysia, Serdang 43400, Malaysia; 4Laboratory of Biocomposite Technology, Institute of Tropical Forestry and Forest Products, Universiti Putra Malaysia, Serdang 43400, Malaysia

**Keywords:** polyethylene, composite materials, ground coffee waste, impact, TGA

## Abstract

The current work discusses ground coffee waste (GCW) reinforced high-density polyethylene (HDPE) composite. GCW underwent two types of treatment (oil extraction, and oil extraction followed by mercerization). The composites were prepared using stacking HDPE film and GCW, followed by hot compression molding with different GCW particle loadings (5%, 10%, 15% and 20%). Particle loadings of 5% and 10% of the treated GCW composites exhibited the optimum level for this particular type of composite, whereby their mechanical and thermal properties were improved compared to untreated GCW composite (UGC). SEM fracture analysis showed better adhesion between HDPE and treated GCW. The FTIR conducted proved the removal of unwanted impurities and reduction in water absorption after the treatment. Specific tensile modulus improved for OGC at 5 vol% particle loading. The highest impact energy absorbed was obtained by OGC with a 16% increment. This lightweight and environmentally friendly composite has potential in high-end packaging, internal automotive parts, lightweight furniture, and other composite engineering applications.

## 1. Introduction

The escalating demand for synthetic fiber has led to over exploration of natural resources causing environmental pollution as they are resistant to degradation. The production and disposal treatment processes of synthetic fiber contribute to environmental pollution as well. In order to protect the earth, natural fiber-reinforced biodegradable polymer matrix and bio-composite have gained attention from researchers worldwide as they are green and eco-friendly materials. Natural fiber is an eco-friendly resource obtained from wood [1], bamboo [2], and agricultural waste [3,4,5,6] such as straw [7] that comprises cellulose, hemicellulose, lignin and aromatics, waxes and other lipids, ash, and water-soluble compounds [8]. It is biodegradable, sustainable, economically viable, low density and possesses good specific strengths and moduli [9,10,11,12,13,14]. Both the natural filler and polymer matrix used in this study are fully biodegradable at the end of their life cycles.

Coffee beans are seeds of the genus flowering plant, Coffea, a member of the botanical family of *Rubiaceace* which comprises almost 500 genera and more than 6500 species [8]. They are normally distributed in tropical and sub-tropical regions. Coffee is the world’s second largest traded commodity, and the top five world leading countries in coffee production are Brazil, Vietnam, Indonesia, Colombia, and Ethiopia [15]. In accordance with the International Coffee Organization, world coffee consumption was about 9 million tons or 149.82 million bags (60 kg per bag) in 2014, an increment of 1.8% since 2013 [16]. Meanwhile, the solid waste generated by coffee production is about one ton annually [17]. Concerted efforts are underway to cultivate sustainable practices for the disposal of residues from processing, such as GCW.

Oxo-biodegradable plastic is a commodity plastic containing highly degradable plastic additive (TDPA) and 2–3% transition metals. The metal salts catalyze the degradation process while speeding it up to degrade abiotically at the end of its useful life in the presence of oxygen faster than ordinary plastic [18,19,20]. The European Standards Organization, CEN, defines oxo-biodegradation as degradation resulting from oxidative and cell-mediated phenomena, either simultaneously or successively [21]. In addition, the useful life of product made using biodegradable plastic could be programmed at manufacture, and the existing technology, machinery and workforce in factories can manufacture it at no extra cost. Biodegradable plastic retains all the advantages of conventional plastic but displays the required additional characteristic of faster degradation after use and disposal by natural mechanisms and with no unwanted environmental consequences [22,23,24].

Fermoso et al. [25] used thermogravimetric analysis (TGA) and mass spectrometry (MS) to carry out pyrolysis tests on spent coffee grounds in a nitrogen atmosphere with heating rates ranging from 5 to 100 °C/min. From the test, they found that the main gases produced during ground coffee pyrolysis are those containing oxygen: H_2_O, CO and CO_2_ within a temperature range between 250 and 425 °C. When a zeolite is used as a catalyst, they found a significant increase in the amount of low molecular weight hydrocarbons such as CH_4_, C_2_H_4_ and C_2_H_6_ [25]. Cai et al. [26] used the thermogravimetric technique together with Fourier transform infrared spectroscopy to analyze the thermal decomposition of some plant residues used in the preparation of tea. They used two mixed N_2_/O_2_ and CO_2_/O_2_ working atmospheres and found that the main gases resulting from the combustion process are: O_2_, H_2_O, CH_4_, CO, CO_2_, NH_3_ and HCN [26]. GCW has a structure that is very similar to components often utilized as filler particles in wood polymer composites. This by-product stands out due to its high protein content (17–18%), which may improve the characteristics of polymer composites. Moreover, because of the presence of these functional groups, it may provide extra opportunities for interfacial bonding within the polar polymeric matrix. Table 1 summarizes the outcomes of previous studies on the mechanical and thermal properties of coffee waste element reinforcement in polymer composites. Even though the hydrophilic nature of GCW may reduce the composite performance due to their poor matrix compatibility and tendency to absorb water, surface treatments may increase the compatibility of the matrix and GCW reinforcement. For example, the treated waste coffee fiber using steam explosion treatment could enhance the thermostability of the composite, increase the fiber crystallinity index and finally be suitable for use as a green composite product [27]. Also, according to Zhao et al. [28], treating the natural fiber using sodium hydroxide (NaOH) increased the interfacial interaction between the matrix and the fiber by up to 300%. Goncalves et al. [29] stated that a NaOH concentration of less than 5 wt% could improve the interfacial bonding between coffee waste and the polyurethane matrix. However, increasing NaOH concentrations up to 10 wt% worsens coffee husk fiber by eroding and deteriorating the structure. Generally, the NaOH solutions ionize the hydroxyl groups in the substrate and chemically alter the fiber structure by dissolving hydrogen bonds. The effectiveness of this process is determined by elements such as response time, heat applied, and solution concentration. 

Furthermore, the proportion of GCW in the structure is an essential component that influences the mechanical performance of composites. Mechanical properties increase if high-strength GCW content increases. However, if the quantity of this GCW exceeds its ideal level, the mechanical capabilities may deteriorate. This is owing to the belief that as fiber volume increases, matrix composition decreases, resulting in poorer interfacial interaction. Table 1 reveals that almost all the investigations used a filler ratio of 1–20 wt% and, in some cases, up to 30 wt% of fiber loading. As can be seen from the table, an improvement of the elastic modulus of coffee silverskin-reinforced PP was obtained at the higher loading of fiber, as reported by Dominici et al. [30]. Similar findings were also reported by Sarasini et al. [31] and Hejna et al. [32]. When coffee silverskin was added to a green polymer, such as poly (butylene succinate) or poly (butylene adipate-co-terephthalate)/poly (3-hydroxybutyrate-co-3-hydroxyvalerate), the tensile strength of the composite structure increased. However, this effect was not significantly improved for stronger poly (lactic acid) matrices [33]. Nevertheless, a great deal of research solely examined the effects of coffee silverskin on morphological, mechanical, and thermal efficiency in composite technology. It was noted that the stiffness and crystallinity of polymer matrices were improved by adding coffee silverskin. However, the effects of GCW on the physico-mechanical properties and thermal stability of composites have not been reported.

**Table 1 polymers-14-04678-t001:** Previous reported work on the mechanical and thermal characteristics of coffee waste-reinforced composites.

Fiber	Matrix	Parametric Evaluation	Particle/ Fiber Loading (wt%.)	Finding	Reference
Coffee chaff and spent ground coffee	PP	Different type of coffee	26	Coffee chaff provided greater thermal stability and was suitable for composite reinforcement.	Zarrinbakhsh et al., 2016 [34]
Coffee silverskin	PBAT-PHBV	Different filler content	10, 20 and 30	Increasing the filler content enhanced Young’s modulus but decreased tensile strength.	Sarasini et al., 2018 [31]
Coffee silverskin	PLA and PBS	Different filler content	10, 20 and 30	Adding fillers to both matrices improved tensile characteristics.	Totaro et al., 2019 [33]
Coffee silverskin	PE	Different filler content	10, 20 and 30	Improvement in the elastic modulus and a reduction in strain at maximum stress were observed with the increase in fiber fraction.	Dominici et al., 2019 [30]
Green coffee cake	PP	Steam explosion treatment	Nil	The addition of treated fiber increased the thermostability of the composite.	*de* Brito et al., 2020 [27]
Coffee silverskin	HDPE	Different filler content	1, 2, 5, 10 and 20	Maximum tensile strength was obtained at 20 wt%. particle loading.	Hejna et al., 2021 [32]
Coffee husk fiber waste	PU	Effect of NaOH concentration	5, 10 and 20	Increased tensile strength by lowering the NaOH content.	Gonçalves et al., 2021 [29]

PP: polypropylene, PBAT: poly(butylene adipate-co-terephthalate), PHBV: poly(3-hydroxybutyrate-co-3-hydroxyvalerate), PLA: poly(lactic acid), PBS: poly(butylene succinate), PE: polyethylene, HDPE: high-density polyethylene, PU: polyurethane.

The aim of this study is to reuse the waste from the coffee beverage industry while reinforcing it with HDPE to produce green and eco-friendly lightweight composites. Oil extracted from GCW using the Soxhlet method can be used to produce biodiesel and for the synthesis of bio-plastic or other potential applications, while the extracted GCW can be used as reinforcement in composite fabrication. The GCW underwent oil extraction and a combination of oil extraction followed by alkaline treatment. The composites were prepared using different particle loadings and were characterized by tensile, impact, FTIR, SEM, TGA, DSC and water absorption.

## 2. Materials and Methods

### 2.1. Materials

Ground coffee waste (GCW) (Arabica) was obtained from a local cafe. The GCW was washed with distilled water, dried in an oven at 105 °C for 24 h to 1–2% moisture content, and then sieved to a particle size of 850 µm. Table 2 shows the chemical composition of GCW from previous work conducted by other researchers. Commercial bio-degradable high-density polyethylene (HDPE) thin film (density of 0.96 g/mL at 23 °C, melt flow index of 0.35 g/10 min, 190 °C/5 kg) was used as the polymer matrix. GCW was extracted using the Soxhlet method in a solvent, n-hexane. Approximately 200 g of dried GCW was extracted in a Soxhlet extractor with 800 ml of n-hexane at a reaction temperature between 80 °C and 95 °C for 3 h. The oil-extracted GCW (OG) was dried at 105 °C for 24 h to evaporate n-hexane. After that, the OG was soaked in 1% concentration of NaOH aqueous solution (0.25 molarity) at room temperature for 24 h. Then, the NaOH-treated OG (ONG) was rinsed with distilled water until it reached pH 7. Both the OG and ONG were dried in an oven at 105 °C for 24 h before use. 

### 2.2. Methods

#### 2.2.1. Compounding and Compression Molding

The dimensions of HDPE thin film used were 230 mm × 230 mm. Each laminate was prepared by sieving GCW in between plies of HDPE films. The compounding of GCW and HDPE were placed in a mold and covered. The composite was prepared using a hot compression molding method at 3 MPa for approximately 15 min until it reached 150 °C. The hot compression mold was then left to cool to room temperature for approximately 2 h before the laminate was taken out. Table 3 shows a summary of all the composites investigated in this research. The results of HDPE and UGC from the previous study were compared, whereby the earlier study performed on modified ground coffee waste showed an enhancement of the properties as compared to untreated ground coffee waste composite [39,40]. OGC is oil-extracted GCW-reinforced HDPE composite (Figure 1) and ONGC is oil-extracted + NaOH GCW-reinforced HDPE composite with different particle loadings.

#### 2.2.2. Composite Property Testing

A.Fourier Transform Infrared Spectroscopy (FT-IR)

FTIR was analyzed using an IRAffinity-1 from Shimadzu, Kyoto, Japan. Approximately 20 mg of GCW each were put on the ATR (Attenuated Total Reflectance). Each sample was subjected to 20 scans in the range of 600 cm^−1^–4000 cm^−1^.

B.Scanning Electron Microscopy (SEM)

SEM (Hitachi TM3030) with 15 kV electrons was used to investigate all the samples. Before observation, the samples were coated with a thin layer of gold using a JEOL JFC-600 auto fine coater.

C.Thermal Analysis
Thermogravimetric Analysis (TGA)TGA was performed via Netzsch TG 200 F3 Tarsus (Selb, Germany). Approximately 20 mg of the composite was prepared and put into a crucible. A temperature range from 30 °C to 600 °C was used with a heating rate of 10 °C min^−1^ in a nitrogen atmosphere. The mass loss, initial degradation temperature (T_onset_), maximum degradation temperature (T_max_), and final degradation temperature (T_final_) were determined.Differential Scanning Calorimetry (DSC)DSC measurements were performed using Perkin Elmer equipment, DSC 8000 model (MA, USA). Approximately 20 mg of the composite was prepared. A temperature range from 50 °C to 180 °C was used with a heating rate of 20 °C min^−1^ in a nitrogen atmosphere. The sample was heated from 50 °C to 180 °C, then maintained at 180 °C for 2 min; the second cycle was performed with a cooling rate of 20 °C min^−1^ until it reached 50 °C; while for the third cycle, the sample was heated from 50 °C to 180 °C at a heating rate of 20 °C min^−1^. The crystalline melting temperature (T_m_) and degree of crystallinity (X_c_) of the HDPE were obtained by considering the second heating curves.

D.Mechanical Properties of the Prepared Composites
Tensile TestingTensile properties were examined using a Shimadzu universal testing machine, model AG-300K IS MS (Kyoto, Japan). Tensile tests were carried out in accordance with ASTM using a load cell of 50 kN at a crosshead rate displacement of 1 mm/min [41]. Tensile strength and tensile modulus were based on initial sample dimensions and the results were averaged over five measurements.Impact TestingDart drop impact properties were characterized using Instron 9250 HV (U.S.) in accordance with ASTM D3763-15. A 12.88 mm hemisphere head of 3.4727 kg was employed and a free-falling initial drop height was set at 0.50 m. A total of five different samples were subjected to impact testing. The composites can be tested to assess their resistance to falling weight [42].

E.Water Absorption

A water absorption test was performed according to ASTM D570-98. Test specimens were prepared at a size of 76.2 mm × 25.4 mm by the thickness of the composite. The specimens were immersed entirely in a container of distilled water maintained at a temperature of 23 ± 1 °C for the test. The specimens were weighed after 2 h and then repeatedly every 24 h after that. The specimens were removed from distilled water one at a time, wiped with a dry cloth and weighed to the nearest 0.001 g. The percentage of water absorption was measured by [43]:(1)$$percentage\;of\;water\;absorption,\;\%=\frac{W_{final}-W_{initial}}{W_{initial}}\times \;100\%$$

## 3. Results and Discussion

### 3.1. FT-IR Spectroscopy

The FT-IR spectroscopy in Figure 2 shows exemplary peaks of untreated GCW (UG), oil-extracted GCW (OG) and oil-extracted + NaOH GCW (ONG), respectively, at 3351 cm^−1^, 2923 cm^−1^, 2859 cm^−1^, 1733 cm^−1^, 1444 cm^−1^, 1367 cm^−1^, 1238 cm^−1^, 1120 cm^−1^ and 1031 cm^−1^. The absorption band of 3351 cm^−1^ belongs to the absorption of –OH stretching vibrations on the GCW. The –OH compound consisted of the presence of water, aliphatic primary and secondary alcohol found in cellulose, hemicellulose, lignin, extractives and carboxylic acids in extractives [44,45]. The decreasing area of this peak represented the removal of lipids. Peaks at 2859 cm^−1^ and 2923 cm^−1^ are recognized as asymmetric and symmetric stretching of C–H bonds in aliphatic chains that are ascribed to lipids [46]. After both treatments, the amount of lipids was reduced. The characteristic GCW peaks at 1733 cm^−1^ and 1238 cm^−1^ were credited to the acetylated xylan (hemicellulose), aldehyde lignin or carbonyl lipid, pectin and wax [47,48]. A peak of 1444 cm^−1^ assigned to benzene ring stretching in lignin and =CH_2_ vibration in polyoses (Hemicellulose) [49], while 1120 cm^−1^ attributed to the aromatic C–H in the plane deformation mode of the guaiacyl/syringyl units of lignin [44] disappeared after both treatments. A peak of 1367 cm^−1^ shown as the in-plane C–H bending in hemicellulose reduced after the treatments [50]. During the 600 cm^−1^–1100 cm^−1^ peak, the –OH concentration increased from UG to OG followed by ONG, revealing this was a more active site available to intermingle between filler/matrix interfaces. Table 4 demonstrates the details of band wavelength and the associated chemical group.

### 3.2. TGA Analysis

Figure 3 shows the emblematic TGA and DTG curve of HDPE, untreated GCW/HDPE (UGC), oil-extracted GCW/HDPE (OGC) and oil-extracted + NaOH/HDPE (ONGC) composites based on particle loadings of 10%. As shown in Table 5 and Table 6, the decomposition began at a lower temperature in the composites than in the neat HDPE due to the presence of GCW. However, the degradation temperature of OGC and ONGC were higher than HDPE at 50% weight loss. The process of oil extractions and alkaline treatment had improved their thermal stabilities. The thermal stability increased in UGC, OGC and ONGC, respectively. The percentage of charred residue decreased from UGC to OGC and followed by ONGC. This showed the treatment effectiveness as more unwanted impurities were removed leading to less charred residue. Weight losses before 100 °C were due to the moisture absorption in the sample. The first step of degradation can be observed in the range of 250 °C–400 °C, marking the degradation of hemicellulose, cellulose and small amount of lignin [51]. The second step of degradation, which is also the main decomposition of GCW, can be observed at approximately 440 °C that contributed to the dramatic degradation of lignin, as shown in DTG curve [52]. The degradation temperature of the materials was increased in the sequence of hemicellulose < cellulose < lignin. 

### 3.3. DSC Analysis

The sample was heated at a heating rate of 20 °C·min^−1^ initially, from 50 °C to 180 °C. When the sample reached 180 °C, it was then left to cool off to 50 °C in approximately 2 min. In the second cycle, the sample was reheated again to 180 °C. The curves exhibited endothermic peaks, as shown in Figure 4. These peaks are attributed to the melting of crystalline domains of the HDPE matrix. Table 7 shows the important data of DSC measurement. Thermal characteristics such as crystalline melting temperature, T_m_, melting enthalpy of composite according to the content of HDPE in the composite, ∆H_m_, and degree of crystallinity, X_c_, were identified. The content of HDPE in the composite, ∆H_m_, was calculated by using the area under the curve; while X_c_ was determined based on the ratio of ∆H_m_ to ∆H^0^_m_, melting enthalpy of 100% crystalline HDPE (293 Jg^−1^) [53]. With regard to melt temperature, the peaks were located in the same temperature range of 137 °C–138 °C. In all composite formulations, X_c_ increased with the incorporation of GCW content. In this case, GCW acted as nucleating agent that produced crystals to increase the percentage of X_c_ in the composites. The nucleating effect was due to the low filler content. Furthermore, the removal of lignin from the GCW after treatment also contributed to an increase in crystallinity [54].

### 3.4. Tensile

The tensile properties were not as good after the inclusion of GCW. Reinforcement with a uniform circular cross-section area and with an aspect ratio more than its critical value would normally improve the tensile strength. However, the irregular shape of GCW fillers with an aspect ratio of less than two cannot support the stress transferred from the polymer matrix [55]. Therefore, the tensile strength reduced significantly. Compared to UGC, both treated GCW/HDPE composites showed an improvement in tensile strength. The tensile strength of the composites was maximized when reinforced with OGC > ONGC > UGC, respectively (Figure 5a). The tensile strength of OGC increased and reached an optimum of 16.47 MPa at a particle loading of 10%, an increment of 27% as compared to UGC, then declined. A large amount of lipids in OG was removed thus exposing a coarser surface with cellulose micro fibrils revealed, as verified by FTIR (Figure 2) and SEM (Figure 10). The exposed cellulose micro fibrils improved the effectiveness of the contact surface area thus enabling better impregnation of GCW by the HDPE matrix. On the other hand, increasing the particle loading led to the decreasing tensile strength of ONGC (Figure 5a). During a particle loading of 5%, the tensile strength was slightly better than UGC at 15.49 MPa, with 8% improvement. This may be due to less exposure of cellulose micro fibrils on its surface after the treatment. In addition, ONG became porous and weaker after alkaline treatment and this led to its having poor properties. 

As observed in Figure 5b, OGC showed a slight improvement as compared to neat HDPE during 5% and 10% of particle volume fraction reaching 1263 MPa (+14%) and 1122 MPa (+1.4%), respectively. The chemical modification proved to provide a better homogeneous dispersion of GCW. The tensile properties showed a proportional increment with particle loadings until 10%. This was due to an enhancement of load transfer between the matrix and GCW interface. Nevertheless, the composites declined in tensile properties beyond optimum values (>10%) due to the agglomeration of GCW. Agglomerations create flaws and voids between GCW and the matrix thus diminishing the tensile properties [56]. Other than that, aspect ratio played a very important role. GCW had a very low aspect ratio of less than two due to insufficient stress transferred; therefore, it displayed a low tensile modulus. In contrast, the tensile modulus obtained for ONGC was lower than HDPE as ONG became weak due to the treatment. 

Figure 5c displays the specific tensile strength, known as strength to weight ratio. All of the treated GCW showed lower specific tensile strength. Particle loadings of 5% and 10% for OGC showed a slight decrease to 0.0173 MPa.m^3^/kg (−1.1%) and 0.0167 MPa·m^3^/kg (−4.6%), respectively. Figure 5d illustrates the specific tensile modulus, known as stiffness to weight ratio. The specific tensile modulus of neat HDPE was 1.0335 MPa·m^3^/kg. The specific tensile modulus improved for every particle loading except for particle loadings of more than 15% for OGC. The specific tensile modulus had improved by 32%. The inclusion of GCW contributed to a lighter weight composite. The specific tensile properties improved slightly as the composites exhibited a lower density. Conversely, the specific tensile properties for ONGC did not improve due to the poor condition of ONG. Another researcher reported the same trend of results using oil extraction oil palm empty fruit brunch composite [57]. Tensile, flexural and toughness properties were improved significantly more than those without extraction.

### 3.5. Impact Test

A dart drop weight impact test was performed to study the impact response of UGC, OGC and ONGC with a 10% particle loading. Figure 6 and Figure 7 show the results. The ascending curve in Figure 6 signifies the stiffness of the composites. The stiffness of the composites increased from UGC, to OGC to ONGC. The maximum impact force corresponds to the onset of material damage. The maximum load was sustained by the composite before fracture increasing in the sequence of UGC < ONGC (+14%) < OGC (+28%), respectively. Meanwhile, the energy to maximum load increased from UGC < ONGC (+11%) < OGC (+51%) as shown by the dotted line in Figure 6. The decreasing curve shows the properties of the materials. The ductility of the composites experienced a similar trend due to the energy of the maximum load. Total energy absorbed increased in the sequence of UGC < ONGC (+7%) < OGC (+16%), respectively. Hydrophilic UG and hydrophobic HDPE had poor interaction thus yielding a lower impact resistance. Both applied treatments had improved impact resistance. After oil extraction, the filler and matrix had a better adhesion and this gave a better uniform distribution of stress transfer. Therefore, more load and energy were required to perforate the samples. The treatments between matrix and reinforcement had significantly improved the toughness of the composite while showing the ability to absorb more energy and deform plastically before fractured. ONGC yielded lower results than OGC. The reason was supported by the results discussed earlier where ONG itself had become weaker. Overall, the impact resistance of the composites enhanced in the sequence of UGC < ONGC < OGC. Figure 8 displays the post-impact fracture behavior of the perforated composites. The failure implicated plastic deformation of the HDPE and debonding of the GCW filler within the composites. With similar particle loading percentages, it seems that the treated GCW composites had better plastic deformation than UGC. The bottom surface of the impacted samples had a bulging effect. Delamination only occurred at the perforated area. A similar impact result trend was obtained by another researcher [58].

### 3.6. Water Absorption

Figure 9 depicts the water absorption proficiency of HDPE, UGC, OGC and ONGC. The water absorption results showed that the non-polar HDPE demonstrated the lowest water absorption. In comparison to similar particle loadings of 10%, the percentage of water absorption increased in HDPE, OGC, ONGC and UGC at 0.05%, 3.22%, 3.80%, 5.22%, respectively, with an improvement of 38% and 27% in UGC. The treated GCW reinforced composites exhibited a lower water absorption rate as it became more hydrophobic after the treatments. Furthermore, a better interfacial adhesion of GCW/HDPE reduced the width of the interface area and diminished water penetration to the material inner parts [59].

### 3.7. Morphological Studies

Figure 10 shows the surface morphology of UG, OG and ONG. All of the GCW particles had a dendritic appearance. The SEM pictograph in Figure 10 illustrates that the modifications conducted on GCW had increased its surface roughness. This had been proven with the FTIR and TGA with the removal of hemicellulose, lignin, pectin, wax, lipid and impurities on the GCW surface. As seen in Figure 10b, a coarser surface was revealed, and cellulose micro fibrils were exposed on the OG surface after extracting the oil. In Figure 10c, ONG showed uneven concave and rougher surfaces with less cellulose micro fibrils exposure. Additionally, ONG was more fragile, as proven by its mechanical properties. 

Figure 11 shows the SEM micrographs conforming to fracture surfaces from the tensile test at 10% particle loading for UGC, OGC and ONGC. For UGC, the fractured surface showed a gap in between the UG and HDPE, which resulted in poor fiber/matrix adhesion. For oil extraction application, a clearer and better interface can be seen between OG and HDPE with a layer of lipid being removed that led to a coarser surface of OG. The adhesion of ONGC was improved to a better impregnation of matrix in ONG. However, ONG became fragile and porous and thus yielded lower mechanical properties.

## 4. Conclusions

The tensile and impact properties, thermal analysis and analytical analysis of untreated and treated GCW/HDPE were evaluated. The characterization results show OGC to be the most impressive among all the fabricated composites. FTIR, SEM and TGA showed the effectiveness of the chemical modification performed on GCW by removing impurities, pectin, wax, hemicellulose, and lignin. Both treatments also produced a more hydrophobic GCW as water absorption rate had decreased. Overall, particle loadings of 5% and 10% exhibited the optimum level for this particular type of composite since mechanical properties had improved as compared to UGC. The incorporation of GCW into biodegradable HDPE aimed to improve the recyclability of waste in order to turn it into a value-added product, minimize ecological damage and reduce the production cost. The composites fabricated are able to degrade themselves at the end of their life cycle without leaving any negative effect on the environment. This particular type of composite has potential for application in consumables, packaging, gardening, furniture parts and automotive components.

## Figures and Tables

**Figure 1 polymers-14-04678-f001:**
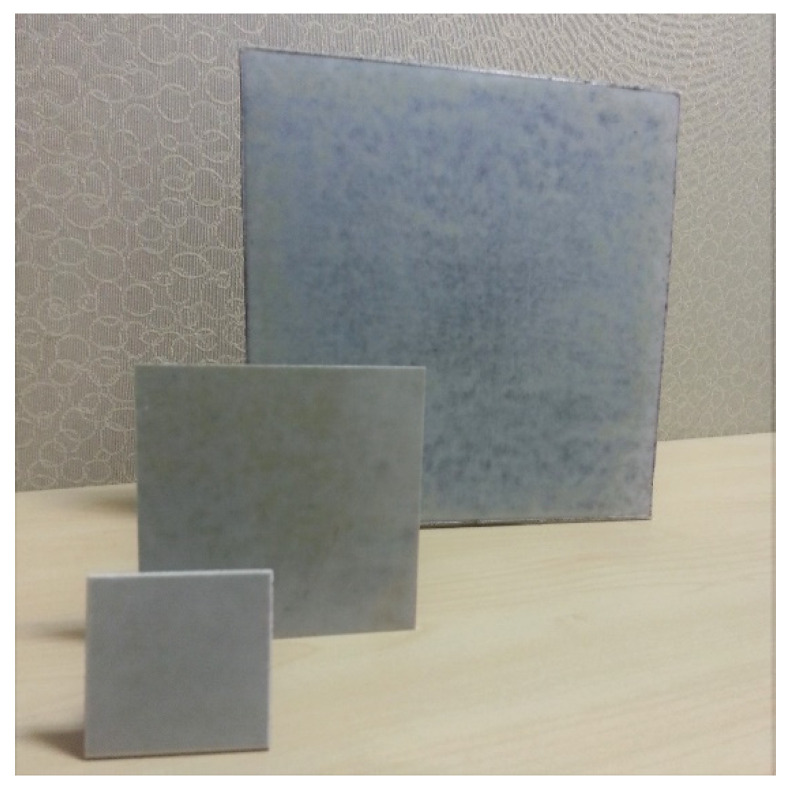
OGC, oil-extracted GCW-reinforced HDPE composite laminate.

**Figure 2 polymers-14-04678-f002:**
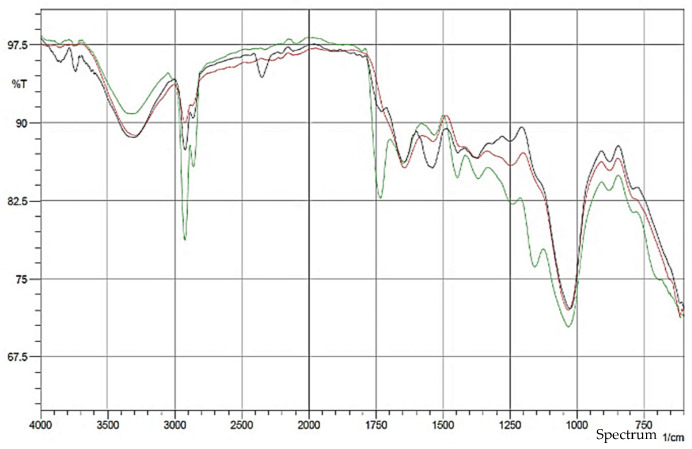
FTIR Spectra of a) UG (green), b) OG (red) and c) ONG (black).

**Figure 3 polymers-14-04678-f003:**
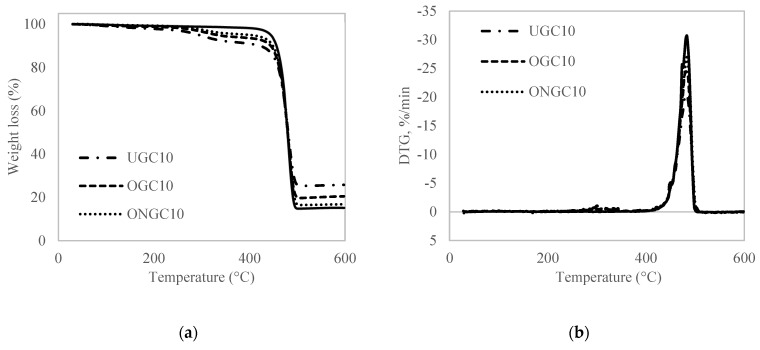
(**a**) Thermogravimetric analysis (TGA) thermogram; and (**b**) Differential thermogravimetric graphics (DTG) thermogram of HDPE (smooth line), UGC, OGC and ONGC (10% particle loading).

**Figure 4 polymers-14-04678-f004:**
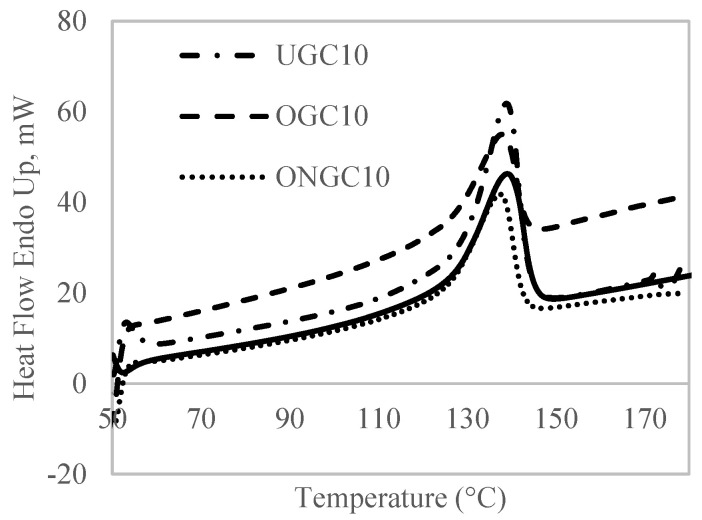
DSC curves of HDPE (smooth line), UGC10, OGC10 and ONGC10.

**Figure 5 polymers-14-04678-f005:**
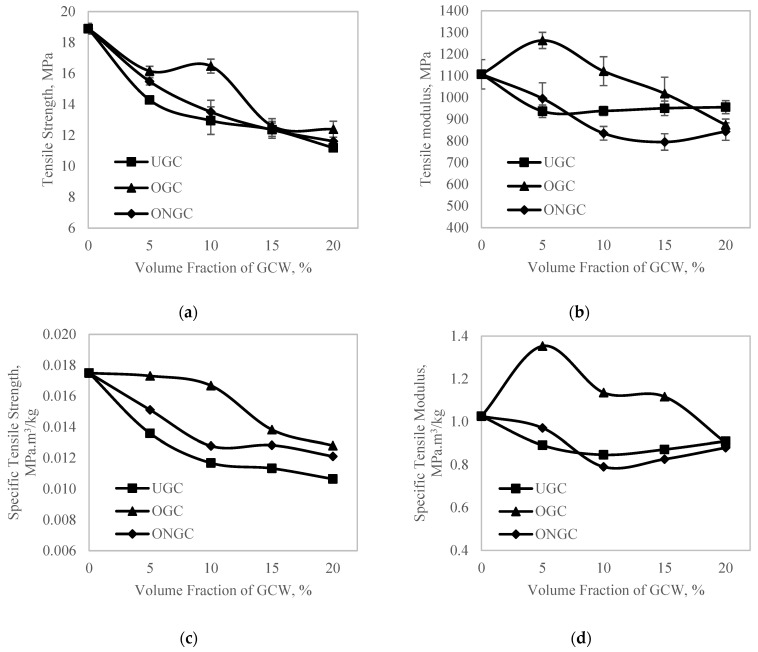
(**a**) Tensile Strength (**b**) Tensile Modulus (**c**) Specific Tensile Strength (**d**) Specific Tensile Modulus of HDPE, UGC, OGC and ONGC of different particle loadings.

**Figure 6 polymers-14-04678-f006:**
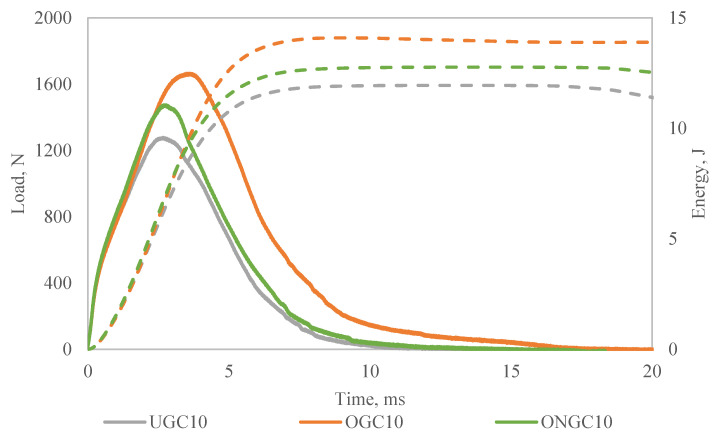
Load-Energy-Time graph of impact test for UGC, OGC and ONGC.

**Figure 7 polymers-14-04678-f007:**
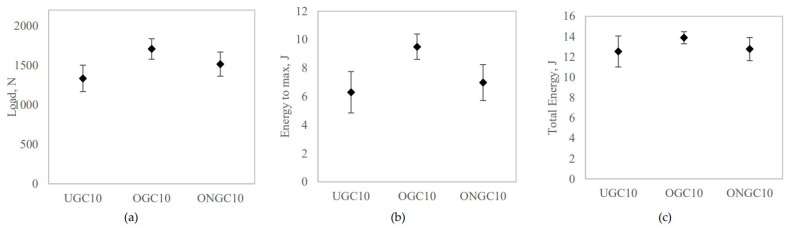
Effect of filler on (**a**) load, (**b**) energy to maximum, (**c**) total energy of GCW/HDPE composite at 10% particle loading.

**Figure 8 polymers-14-04678-f008:**
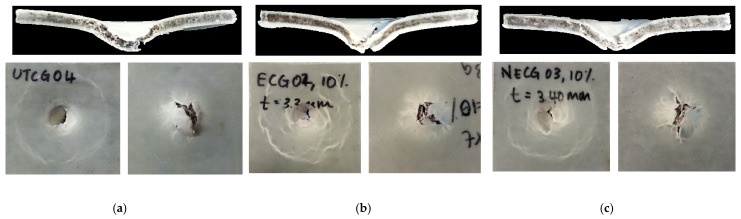
Photograph of fracture behavior of perforated samples (**a**) UGC, (**b**) OGC, (**c**) ONGC (10% particle loading).

**Figure 9 polymers-14-04678-f009:**
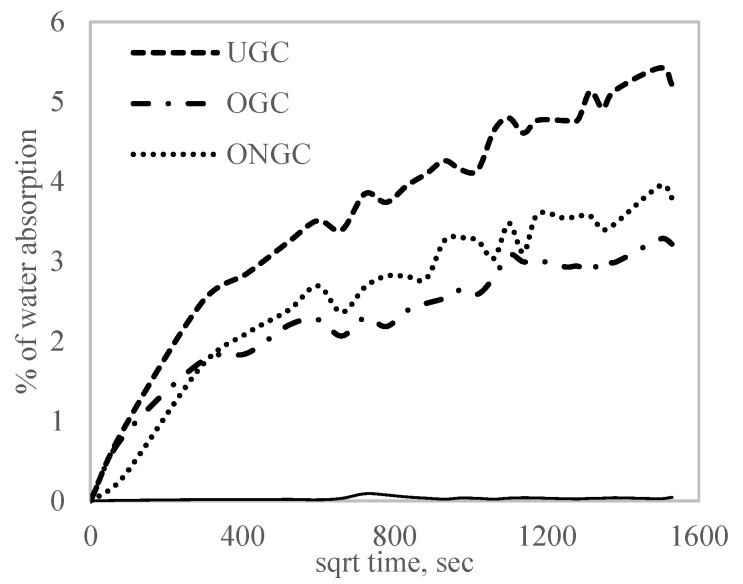
The water absorption proficiency percent mass gain for all the composites with 10% particle loading in comparison with HDPE (smooth line).

**Figure 10 polymers-14-04678-f010:**
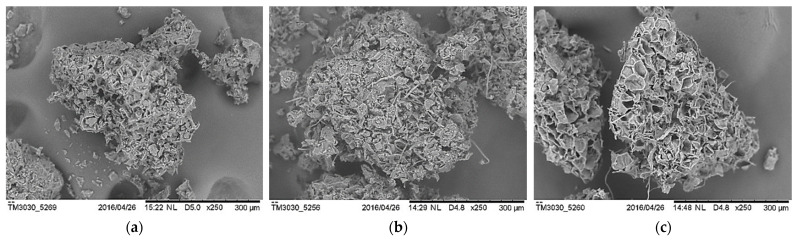
SEM micrograph of GCW particles of (**a**) UG, (**b**) OG and (**c**) ONG.

**Figure 11 polymers-14-04678-f011:**
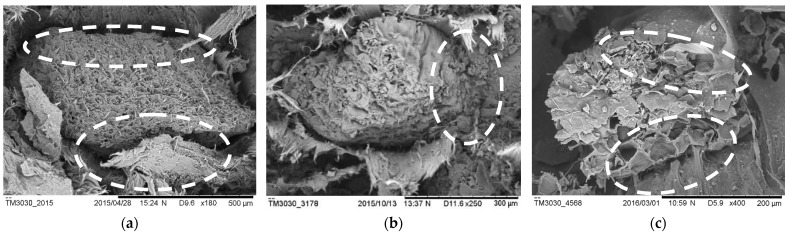
SEM micrograph of breaking surface of (**a**) UGC, (**b**) OGC, (**c**) ONGC (10% particle loading).

**Table 2 polymers-14-04678-t002:** Chemical composition of GCW.

Composition of GCW (%)			Reference
Cellulose	Hemicellulose	Lignin	
13.02	-	26.52	[35]
13.85	-	19.84	[35]
13.8	-	33.6	[36]
12.4 ± 0.79	39.1 ± 1.94	23.9 ± 1.7	[37]
8.6	36.7	-	[38]

**Table 3 polymers-14-04678-t003:** Summary of type of composites investigated in this study.

Composite	Sample	Stacking Sequence	Volume of GCW (%)	Volume of HDPE (%)	Reference
HDPE	HDPE	150PE	0	100	[39]
UGC5	Untreated GCW5/HDPE	150PE + 5UG	5	95	[39]
UGC10	Untreated GCW10/HDPE	150PE + 10UG	10	90	[39]
UGC15	Untreated GCW15/HDPE	150PE + 15UG	15	85	[39]
UGC20	Untreated GCW20/HDPE	150PE + 20UG	20	80	[39]
OGC5	Oil Extraction GCW5/HDPE	150PE + 5OG	5	95	
OGC10	Oil Extraction GCW10/HDPE	150PE + 10OG	10	90	
OGC15	Oil Extraction GCW15/HDPE	150PE + 15OG	15	85	
OGC20	Oil Extraction GCW20/HDPE	150PE + 20OG	20	80	
ONGC5	Oil Extraction + NaOH5/HDPE	150PE + 5ONG	5	95	
ONGC10	Oil Extraction + NaOH10/HDPE	150PE + 10ONG	10	90	
ONGC15	Oil Extraction + NaOH15/HDPE	150PE + 15ONG	15	85	
ONGC20	Oil Extraction + NaOH20/HDPE	150PE + 20ONG	20	80	

UG = Untreated ground coffee waste, OG = Oil-extracted ground coffee waste, ONG = Oil-extracted + NaOH ground coffee waste.

**Table 4 polymers-14-04678-t004:** FTIR results of UG, OG and ONG.

Band Wavelength (cm^−1^)	Associated Chemical Group	UG	OG	ONG
3351	–OH intensity	High	Low	Low
2923	Lignin peak	Present	Reduced	Reduced
2859	Lignin peak	Present	Reduced	Reduced
1733	Hemicellulose, Lignin, Lipid, pectin-wax peak	Present	-	-
1444	Lignin, hemicellulose peak	Present	-	-
1367	Hemicellulose peak	Present	Reduced	Reduced
1238	Hemicellulose, Lignin peak	Present	Reduced	Reduced
1120	Lignin peak	Present	-	-
600–1100	–OH intensity	Low	High	High

**Table 5 polymers-14-04678-t005:** Summary of TGA data for HDPE, UGC, OGC and ONGC (10% particle loading).

Composites	T_onset_ (°C)	T_max_ (°C)	T_final_ (°C)	Residue (%)	Reference
HDPE	461.8	474.4	493.2	15.13	[39]
UGC10	460.4	481.9	492.4	25.80	[39]
OGC10	460.4	481.9	492.4	20.47	
ONGC10	459.1	482.0	492.8	17.71	

**Table 6 polymers-14-04678-t006:** Degradation temperature of HDPE, UGC, OGC and ONGC (10% particle loading).

Composites	Temperature at 10% Weight Loss (°C)	Temperature at 15% Weight Loss (°C)	Temperature at 25% Weight Loss (°C)	Temperature at 50% Weight Loss (°C)
HDPE	456.52	462.48	469.74	479.60
UGC10	422.11	450.39	464.64	478.01
OGC10	444.50	456.08	466.70	479.88
ONGC10	448.79	458.33	467.87	480.29

**Table 7 polymers-14-04678-t007:** Summary of DSC data of HDPE, UGC10, OGC10 and ONGC10.

Composites	T_m_ (°C)	∆H_m_ (J/g)	X_c_ (%)
HDPE	138.92	142.44	48.61
UGC10	138.85	133.27	50.54
OGC10	137.72	140.67	53.34
ONG10	137.35	140.76	53.38

## Data Availability

Not applicable.

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
