# Peer review of "Evaluation of Physico-Mechanical Properties on Oil Extracted Ground Coffee Waste Reinforced Polyethylene Composite"

_polymers, 2022, doi:10.3390/polym14214678_

Round 1

Reviewer 1 Report

The paper presents the effect of the content and the possibility of surface modification of used coffee grounds as a filler in the HDPE matrix.

The topic is very interesting and valuable, but I have a few questions / suggestions:

1. Literature introduction is poor in information. I suggest extending the literature introduction on the possibility of using other biodegradable polymers, e.g. PHBV with fillers of plant-delivered (you can use the publication from the POLYMERS journal)

2. The discussion is poor in information on the current state of knowledge. I propose to extend the discussion to the literature (e.g. from the POLYMERS journal) with the current state of  the alkali treatment of plant-delivered with NaOH solution and the influence on the properties of biocomposites.

3. Have other NaOH concentrations been used? Why was 1% chosen exactly?

4. Why was 5, 10, 15, 20% filler content chosen. Has the maximum filler content in the HDPEmatrix been determined, where the resulting composite does not delaminate?

5. I recommend that you include photos of the obtained composites (at least one example) in the work to show the distribution of the filler in the polymer matrix and to refer to, inter alia, for phase homogenization.

6. Has the phenomenon of sticking of the filler particles been noted? If so, has the application of surface modification reduced the tendency of coffee grounds to stick together?

Please respond to the comments and make corrections as suggested.

Author Response

Dear Reviewer, 

Thank you very much for the useful comments. We have made a good effort to improve the manuscript thoroughly and accordingly. Please refer to the attachment for the response of your comments. 

Reviewer 2 Report

Dear,

The authors produced polyethylene composites with coffee powder waste. The authors need to show the importance of reusing this residue for the production of composites and, at the same time, the potential of the results obtained. In addition, some improvements are listed below:

> Abstract.

The main results of the composites should be presented in the abstract, especially with percentage gains. The authors must present a final conclusion about the composites. Does coffee waste have potential as filler or reinforcement? This should be addressed in the abstract;

“.....followed by a hot compression molding with different GCW particle loading (5%, 10%, 17 15% and 20%”. The correct term would not be filling instead of loading? Please review the manuscript;

> Introduction.

Authors need to clarify the novelty of the manuscript, since there are works in the literature on the subject.

The authors need to add a specific review on polymer composites with coffee powder;

In my opinion, the authors should address the importance of reusing coffee powder as a natural filler for the production of composites. In addition, comment on the importance of reintroducing the production chain to produce ecological materials and add value to discarded material;

The subject of ecological composites is current, but the authors used many old references. Therefore, a renewal of references is recommended, especially between 2018-2022.

> Materials

“Ground coffee waste (GCW) (Arabica) was obtained from local cafe. The GCW was 68 washed, dried and then sieved to particle size of 850 µm”. Please inform the drying temperature; oven with vacuum or air circulation? washed with distilled water? Informs the details;

“high-density polyethylene (HDPE)...........”. Inform the commercial code; inform the melt flow rate (MFR);

> Compounding and compression molding

How were the composites mixed to obtain good filler dispersion in the PE matrix? In an extruder? Internal mixer? mixed thermokinetic? Were the materials dried before compression molding? What was the cooling time of the compression molded samples? Please add experimental details;

> FTIR. Inform the resolution used. How thick is the sample? Was it powder? Thick sample? film?

> TG and DSC. Inform the mass used for each analysis. Informs the flow of gas used;

> Tensile testing. Please inform the load used in the equipment;

> Please add the name on the "x" and "y" axis of Figure 1;

> The authors need to add the detailed characterization of the coffee powder: TG; FTIR; granulometric distribution

> Why did the authors not add the crystallization temperature curves? This is important to assess whether there has been a shift to higher temperature. Are the results melting temperature from the first run or from the second?

> In general, the results must be compared with other similar systems in the literature. In addition, the authors must show the importance of the results, since they use a waste;

Author Response

(The authors gave the same response as above.)

Round 2

Reviewer 1 Report

Thank you for correcting the publication and for providing comprehensive answers to the questions. The manuscript is ready for publication.

Reviewer 2 Report

The authors accommodated the recommendations in the manuscript, improving quality. In view of this, the manuscript has merit for publication.